# Repercussions faced by health professionals who have experienced the phenomenon of the second victims due to incidents related to patient safety: A scoping review protocol

Elaine Cristine da Conceição Vianna Gonçalves da Costa[1*],
Luana Cardoso Pestana[1], Laríssia Admá de Souza Pereira[1],
Fabrício Glauber Suzano Maciel[1], Cíntia Silva Fassarella[1],
Renata Eloah de Lucena Ferretti-Rebustini[2], Cristiane Helena Gallasch[1]

**1** Nursing School. Post-Graduation Program in Nursing. Universidade do Estado do Rio de Janeiro/Rio de Janeiro, Rio de Janeiro, Brazil, **2** Nursing School. Postgraduate Program in Nursing in Adult Health. University of São Paulo/ São Paulo, São Paulo, Brazil

☺ These authors also contributed to this work.
\* enfavianna@gmail.com.br

## Abstract

### Introduction:

Patient safety is a relevant, timely and globally significant topic. In hospital settings, despite ongoing discussions and advances in patient safety, adverse events continue to occur, impacting not only patients but also healthcare professionals. Health professionals involved in adverse events are considered second victims. This justifies the growing and pertinent interest in the phenomenon of second victim, as this experience affects the competencies and skills of health professionals while influencing their personal livesand general health.

### Objective:

To map and characterize the repercussions experienced by health professionals who have experienced the phenomenon of second victims due to incidents or adverse events related to patient safety in health services.

### Method:

We will conduct a scoping review following the Joanna Briggs Institute's methodology for scoping reviews, in alignment with the guidelines of the Preferred Reporting Items for Systematic Reviews and Meta-Analyses for Scoping Reviews (PRISMA-ScR) extension. The databases to be searched include PUBMED, SCOPUS, CINAHL, Web of Science (WOS) and LILACS, while the gray literature will be retrieved from the Brazilian Digital Library of Theses and Dissertations (BDTD) and PROQUEST.

**Data availability statement:** The scoping review protocol is registered in the Open Science Framework (DOI: https://doi.org/10.17605/OSF.IO/TG73S). No datasets were generated or analyzed during the current study. All relevant data from this study will be made available upon completion of the study.

**Funding:** This proposal was financed in part by the Coordenação de Aperfeiçoamento de Pessoal de Nível Superior – Brazil (CAPES) – Financing Code 001.

**Competing interests:** The authors declared that there are no competing interests.

The identified studies will be compiled and uploaded to EndNote (version X9.3.3), where duplicates will be removed. Two independent reviewers will import the studies into the Rayyan QCR® for organization and selection, with disagreements resolved by a third party. The selection of studies will be based on the population, the concept investigated and the context. The data will be extracted through a pre-established form.

## Results:

The findings will be analyzed in a descriptive and narrative way, aligning the data with the objective of the study and the research question. The data will are presented in tables, diagrams, and graphs containing information from the extraction tool.

## Introduction

The topic of patient safety is relevant, timeless and of global interest, and although changes and advances in discussions about patient safety are driven, adverse events still occur, causing victims that go beyond the patient [1–3].

An adverse event is defined as an unintentional incident with physical, psychological, or social harm to the patient resulting from a medical error, failure in the health system, or side effects of a medication or treatment1 [2,4].Undoubtedly, adverse events (AEs) are devastating and directly affect patients, who are considered the first victims. In addition to patients, health professionals involved in AEs also suffer, becoming the second victims [4].

The term second victim was first coined by researcher Albert Wu in 2000 [5] to describe medical professionals who made mistakes and became victims. In 2009, the concept was expanded to include all health professionals who had experienced psychological and professional suffering after an adverse event, medical error, or unforeseen injury caused to the patient [2,6].

The Global Action Plan for Patient Safety 2021–2030 increases that the safety of healthcare professionals and patients consists in inseparable and connected domains, with one directly directly impacting the other, and that risks to the health and safety of workers can lead to risks, harms, and adverse events for patients [7].

Considering that events, related to the phenomenon of second victim (PSV), impact in private life, relationships with coworkers and the social context of these professionals, studies that identify, in addition to symptoms, the consequences of PSV among health professionals involved in incident events or adverse events related to health care are justified [1,2,8].Thus, to identify the existence of publications or scoping review protocols registered on the repercussions of the second victim phenomenon among health professionals, a preliminary study was carried out in the Virtual Health Library (VHL), the Medical Literature Analysis and Retrieval System (MEDLINE) and the Cumulative Index to Nursing and Allied Health Literature (CINAHL), as well as the Open Science Framework (OSF), PROSPERO, and the Joanna Briggs Institute's (JBI) Evidence Synthesis platforms.

This preliminary search identified 15 studies, including systematic reviews, scoping reviews, scoping review protocols, and registered scoping reviews [9–23].

Among these publications, two scope reviews [9–10] aimed to map and analyze knowledge about strategies to support health professionals who experience the experience of the second victims (EV). These reviews also examined the Second Victim Experience and Support Tool (SVEST).

The studies included in the systematic reviews analyzed coping and support strategies to address the PSV among health professionals, to define concepts, and to describe the prevalence of psychological and psychosomatic symptoms among second victims [11–16].

On the PROSPERO platform, we found a systematic review protocol registered in March 2024, addressing research trends related to the phenomenon of the second victim [17] In the JBI database, we identified two registries: a protocol and an article with the aim of synthesizing the best available evidence to explore the meaning of nurses' experiences as second victims of adverse nursing errors [19–20].

Finally, we found four registers on the OSF [20–23] platform, including one registered protocol and three research projects, one of them closely aligned with this proposal, entitled "Psychological and physical symptoms of health professionals after a safety incident and understanding their social and professional impacts: a scoping review protocol" [23]. However, that protocol limited the search to English and Finnish studies conducted between 2020 and 2024. In this study, the aim is to broadly identify repercussions of PSV without language restrictions or time frames.

Among the 15 identified [9–23], two are closely aligned with the objective of this review protocol. However, despite addressing the impact of adverse events on second victims, those publications present limitations. Review studies are not contemporary. In addition, health systems and work environments have undergone relevant epidemiological and technological transformations since the publication, which has different implications for health professionals. Finally, these studies present gaps for current research, such as the systematic review publication conducted in 2013 addressing of the scope review for a specific population (surgeons). Considering the significant changes in the labor context and health organizations, particularly with the advancement of patient safety and given the lapse of time since the last review, we intend to contribute to updating and identifying the current repercussions of the second victim phenomenon on health professionals [1,6–8].This scoping review protocol aims to map and characterize the repercussions suffered by health professionals who experienced the phenomenon of second victim resulted from incidents and adverse events related to patient care in health services.

## Method

### Study design

The scoping review approach is considered appropriate for gathering several types of evidence, addressing the complexity of the study's objective, and allowing a broad, exploratory, and descriptive mapping of the repercussions of the second victim phenomenon. We will conduct the scoping review following the JBI methodology for scoping reviews [24], aligning with the Preferred Reporting Items for Systematic Reviews and Meta-Analyses for Scoping Reviews (PRISMA-ScR) extension [25]. No restrictions will be applied regarding study design, year of publication, or language, ensuring access to relevant national and international literature on the research question.

The current protocol follows the guidelines of the Preferred Reporting Items for Systematic Review and Meta-Analysis Protocols (PRISMA-P) [26] and was registered in the Open Science Framework in https://doi.org/10.17605/OSF.IO/TG73S.Based on the mnemonic PCC [27], the population (P) will be composed of health professionals, the concept (C) investigated will be the phenomenon of the second victim and the context (C) will cover incidents related to patient safety in health services. The review question is: **What does the literature report about the repercussions experienced by health professionals who experience the phenomenon of the second victim due to incidents and adverse events related to patient safety in health services?**

In addition, this study aims to identify answers to the following primary questions:

• What tools are available to identify the phenomenon of the second victim among health professionals?

• What tools exist to assist, manage or support the professionals involved in the phenomenon of the second victim?

• What types of patient safety-related incidents are reported in the included studies?

**Eligibility criteria**

Given that a scoping review can include all types of literature [24–28], this study will consider primary studies and reviews identified in the databases, as long as they have repercussions resulting from the phenomenon of second victims in health professionals.

We will use the Brazilian Digital Library of Theses and Dissertations (BDTB) and Proquest Dissertations and Theses Global as source for gray literature.

Studies that meet the PCC criteria will be included if the participants were [28,29] higher level or licensed health professionals or mid-level or unlicensed professionals. Examples of higher education or graduate professionals include nurses, physicians, physical therapists, speech therapists, and nutritionists, among others. Mid-level or nonregulated professionals include nursing technicians, laboratory technicians, and radiology technicians, among others.

The concept investigated will be the phenomenon of the second victim, which refers to the experience of health professionals involved in an unexpected adverse event, medical error, and/or an injury related to the patient, when they become victims and develop trauma related to that event [3–5].

Regarding the context of the study, we will consider health services based on the scope of Health Sciences Descriptors (DeCS) and Medical Subject Headings (MESH)[30], where health services are defined as an organized system for the provision of health care within a country. The range of services can vary depending on the country, covering preventive services as well as care [30].

**Exclusion criteria**

Abstracts of scientific events, opinion articles, and publications that did not meet the PCC and the objective of this review were not included in the sample. Studies on the adaptation and validation of instruments, as well as studies on the prevalence of symptoms, studies about types of coping and support that did not show the repercussions were disregarded in the selection. In addition, publications that were not available in full text, as well as those that exclusively addressed students from regulated or nonregulated health professions, were excluded.

**Search strategy**

The search strategy will be carried out in three stages, according to the recommendations of JBI methodology [24].

The first stage was carried out on January 8, 2025, with a search limited to the MEDLINE database (PUBMED), using a combination of descriptors with adherence to the review. At this time, 371 publications were recovered, and the strategy for the second stage was customized (Table 1).

In the second step, an experienced and specialized librarian in scoping review research strategies will conduct a comprehensive literature search in the PubMed, Scopus, Web of Science (WOS), CINAHL, and LILACS databases and by the gray literature in the dissertation and thesis database (BDTD) and Proquest Dissertations and Theses Global. The librarian will correlate controlled descriptors and free-text synonyms related to the PCC mnemonic to identify scientific evidence that answers the research question of this study. To ensure the accuracy of the search terms, we will consult sources of controlled vocabulary in health, including the Health Sciences Descriptors (DeCS) and Medical Subject Headings (MeSH).

In the third step, we will examine the reference lists of the selected texts to decide which publications will meet the inclusion criteria to be included.

**Table 1. Details of the initial search carried out in the first stage of the search strategy for scoping review according to JBI recommendations.**

| Consulted | Term mapping | Records Retrieved |
|---|---|---|
| #1 | (second victim phenomenon) OR (or)) OR (second victim) | 3,666 |
| #2 | ((Healthcare Personnel) OR (Healthcare Professionals)) OU (Health Professional) | 888,904 |
| #3 | (Health Bureau) | 3.264,147 |
| #4 | #1 & #2 & #3 | 371,063 |

Source: Author, Rio de Janeiro, Brazil, 2025.

## Screening and selection of evidence

Initially, all identified records will be compiled and uploaded to EndNote (version X9.3.3), where duplicates will be removed.

The corresponding researcher will lead the pilot phase of the study selection process. First, a pilot test will be conducted with the selection team to ensure that all reviewers are familiar with the inclusion and exclusion criteria. This phase will also allow for the necessary refinements. To begin the screening process, both reviewers will conduct a preliminary evaluation of 25 titles and abstracts.

Two independent reviewers will select titles and abstracts, read full texts, and exclude studies using the Rayyan QCR® [30–32] platform to organize manuscripts. If disagreements arise at any stage of the selection process, a third reviewer will solve the conflict.

Concerning the language of the selected articles, the reviewers are proficient in reading manuscripts in English and Spanish. If necessary, for studies published in other languages, a translation app or a professional translator will be used. We will document the use of these mechanisms in the final paper.

The selection of studies will follow the inclusion criteria established in the PCC framework, and the results will be present in a compatible PRISMA-ScR flowchart report.

## Data extraction

Data will be extracted from the selected articles to present the scoping review. To do so, reviewers will use the extraction form presented in Table 2, which details the material analyzed.

The draft version of the data extraction tool can be modified and revised as needed throughout the extraction process. If any modifications occur, it will be document in the final scoping review.

If necessary, authors of the selected publications can be contacted twice to request additional information about the articles or to obtain the full manuscript.

## Analysis and presentation of data

We will analyze the results of the scoping review performing a descriptive and narrative analysis, connecting the obtained data to the aim of the study and the research question. The data found will be presented in tables, diagrams and graphs, presenting the extracted information.

To illustrate the findings about the reported repercussions, we also plan to create a concept map and/or word cloud.

## Expected results

We aim to map and characterize the repercussions experienced by health professionals as result of the PSV in all its dimensions.

In addition to recognizing that the phenomenon of the second victim causes psychic suffering [1–4]. It is essential to characterize how this suffering manifests itself and what repercussions it can lead to.

**Table 2. Data extraction form with information description.**

| Data extraction form following JBI recommendations for scoping review | |
| --- | --- |
| Repercussions faced by health professionals who have experienced the phenomenon of the second victims due to incidents related to patient safety: a scoping review protocol | |
| Authors: Elaine Cristine da Conceição Vianna Gonçalves da Costa, Luana Cardoso Pestana, Laríssia Admá de Souza Pereira, Fabrício Glauber Suzano Maciel, Cíntia Silva Fassarella, Renata Eloah de Lucena Ferretti-Rebustini, Cristiane Helena Gallasch | |
| Year of completion of the review 2025 | |
| Extracted information | Description |
| Title | Study title |
| Authors | Full names of all authors |
| Year | Year of publication |
| Country | Country where the study was conducted |
| Publication type | Type of article, thesis, dissertation, report or guideline |
| Goal | Description of the purpose of the study |
| Study design | Identification of the study design as described by the author |
| Population | Indication of which professionals participated in the study |
| Sample | Number of survey participants |
| Study environment | Type of health service |
| Definition | Key concepts used in relation to the phenomenon of the second victim |
| Findings | Description of the main results of the study |
| Reported repercussions | Description of the reported repercussions due to the phenomenon of the second victim |
| Identification tools | Identification of instruments used to recognize the phenomenon of the second victim |
| Support Tools | Types of assistance, interventions or support provided to professionals affected by the phenomenon of the second victim |

Source: Author, Rio de Janeiro, Brazil, 2025.

Thus, results will be categorized into three thematic areas on the basic of the selected publications: physical repercussions, mental repercussions, and psychosocial repercussions.

We will analyze and discuss these categories from the perspective of occupational health, identifying vulnerabilities in the workplace that may contribute to patient safety incidents and affect the ability of professionals to address them.

In addition, at the national and international level, it is expected that results provide evidence characterizing those repercussions, fostering health policies and practices that ensure both patient safety and the well-being of the professionals involved in these incidents.

## Limitations of the study

The expression "second victim" is not a standardized and vocabulary in DeCS/MESH, preventing the use of synonyms for this concept in the bibliographic search.

## Supporting information

**S1 Checklist.**
(PDF)

## Author contributions

**Conceptualization:** Elaine Cristine da Conceição Vianna Gonçalves da Costa, Cíntia Silva Fassarella.

**Data curation:** Elaine Cristine da Conceição Vianna Gonçalves da Costa, Cristiane Helena Gallasch.

**Formal analysis:** Elaine Cristine da Conceição Vianna Gonçalves da Costa, Cíntia Silva Fassarella, Renata Eloah de Lucena Ferretti-Rebustini, Cristiane Helena Gallasch.

**Funding acquisition:** Elaine Cristine da Conceição Vianna Gonçalves da Costa.

**Investigation:** Elaine Cristine da Conceição Vianna Gonçalves da Costa, Luana Cardoso Pestana, Laríssia Admá de Souza Pereira, Fabrício Glauber Suzano Maciel.

**Methodology:** Elaine Cristine da Conceição Vianna Gonçalves da Costa, Luana Cardoso Pestana, Cíntia Silva Fassarella, Cristiane Helena Gallasch.

**Supervision:** Cristiane Helena Gallasch.

**Validation:** Cíntia Silva Fassarella, Renata Eloah de Lucena Ferretti-Rebustini, Cristiane Helena Gallasch.

**Visualization:** Elaine Cristine da Conceição Vianna Gonçalves da Costa, Luana Cardoso Pestana, Renata Eloah de Lucena Ferretti-Rebustini.

**Writing – original draft:** Elaine Cristine da Conceição Vianna Gonçalves da Costa.

**Writing – review & editing:** Elaine Cristine da Conceição Vianna Gonçalves da Costa, Luana Cardoso Pestana, Laríssia Admá de Souza Pereira, Fabrício Glauber Suzano Maciel, Cíntia Silva Fassarella, Renata Eloah de Lucena Ferretti-Rebustini, Cristiane Helena Gallasch.

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
