## [Decision Letter · Decision Letter 0]

Dear Dr. Vianna Gonçalves da Costa,

Thank you for submitting your manuscript to PLOS ONE. After careful consideration, we feel that it has merit but does not fully meet PLOS ONE’s publication criteria as it currently stands. Therefore, we invite you to submit a revised version of the manuscript that addresses the points raised during the review process.

We look forward to receiving your revised manuscript.

Kind regards,

Vincenzo De Luca

Academic Editor

PLOS ONE

2. Thank you for stating the following financial disclosure:  [This proposal was financed in part by the Coordenação de Aperfeiçoamento de Pessoal de Nível Superior – Brazil (CAPES) – Financing Code 001.]. 

Additional Editor Comments (if provided):

Reviewers' comments:

Reviewer's Responses to Questions

**Comments to the Author**

1. Does the manuscript provide a valid rationale for the proposed study, with clearly identified and justified research questions?

Reviewer #1: Yes

2. Is the protocol technically sound and planned in a manner that will lead to a meaningful outcome and allow testing the stated hypotheses?

Reviewer #1: No

3. Is the methodology feasible and described in sufficient detail to allow the work to be replicable?

Reviewer #1: No

4. Have the authors described where all data underlying the findings will be made available when the study is complete?

Reviewer #1: Yes

5. Is the manuscript presented in an intelligible fashion and written in standard English?

Reviewer #1: No

You may also provide optional suggestions and comments to authors that they might find helpful in planning their study.

Reviewer #1: Thank you for the opportunity to review this scoping review protocol manuscript. This is an interesting topic in the field of patient safety. However, the protocol requires additional revisions. Below are comments for your consideration:

1) Ensure consistency in your writing mechanics – you use past/present/future tense in multiple sections which detracts from your main messages. There are also issues with sentence structure, typos and grammatical issues that should be addressed to improve flow and readability.

2) Abstract - Clarify the abstract. Some sections are unclear. For example, you state “Despite these advances, adverse events victims, including professionals and institutions, to progress beyond the patient.” It is not clear what this sentence is conveying.

3) Introduction - The introduction requires significant revisions as it was difficult to comprehend. Particular attention is needed to improve sentence/paragraph structure, organization and flow. Consider strengthening the background to build the case for why this review is needed and reduce the amount of content dedicated to explaining existing systematic/scoping reviews. The main purpose is to demonstrate that your proposed scoping review is different from what already exists in the literature.

4) Global approach - You note that your scoping review will take a global approach; however, you begin your introduction by referencing the Brazilian National Program for Patient Safety, which is very specific to one country. Consider integrating literature from a global perspective.

5) Methodology- Please provide a rationale for why the scoping review methodology (compared to other literature review methods) is the best choice for answering your questions.

6) Review question - Your review question is: “What does the literature present about the phenomenon of second victim and the repercussions suffered by health professionals involved in incidents related to patient safety in health services”. However, in your review of the literature (on page 7) you note that there are already systematic reviews looking at the impact of adverse events on second victims. Please further clarify how your scoping review will make a different contribution to the literature.

7) Language - You note that you will not limit the language – please indicate how you will review sources that are not in English or in the languages that your team is proficient in.

8) Population/participants – you separate health professionals into two categories – higher education and secondary education. Please clarify this as this conceptualization may be well known across global jurisdictions. One suggestion is to state that you include both regulated and non-regulated health professionals.

9) Line 219 is unclear “the choice of the term “health professionals” instead of “health workers” is based on the definition that a health professional is a person who works in one of the professions related to health sciences, different from health professionals’ health, which includes…”. This sentence seems to be missing something.

10) Line 245 subheading – please clarify what is ‘types of fonts’. You may be referring to ‘types of sources’

11) You noted that a full search was already completed on June 26th – please provide the full search as an appendix. Further, has this review already started? If the full search was conducted in June 2024, it may be worth updating the search before you commence the review to ensure the most up-to-date literature is included.

12) Sources- Please provide further information about the types of sources. For example, what type of studies (Qualitative, quantitative, mixed methods) will you include? Will you include non-empirical sources? What sources will you retrieve grey literature from? Why? what is your rationale for including such sources?

13) Selection of studies – please clarify if you will be piloting the process of selecting studies to ensure members of your team are clear about the inclusion and exclusion criteria and to determine if refinements are needed.

14) Data extraction – please clarify what you mean by 75% agreement (line 291-292)

15) Expected outcomes – please expand on what you anticipate to find and why. Please also clarify what you mean by “this research can contribute to the mapping of the theoretical framework on the phenomenon… (lines 326-328). Is this an existing framework?

**Do you want your identity to be public for this peer review?** For information about this choice, including consent withdrawal, please see our Privacy Policy

Reviewer #1: No

---

## [Author Response · Author response to Decision Letter 1]

19 Feb 2025

Dear all,

The manuscript was adapted in terms of title, symbols and author participation, following the indicated model.

Regarding funding, I would like to inform you that the Coordination for the Improvement of Higher Education Personnel (CAPES) had no role in the study design, data collection and analysis, decision to publish or preparation of the manuscript. This funding agency represents a government agency that funds the corresponding author's doctoral scholarship with funding code 001 (social demand).

Regarding the study design, it is not clear to the authors where there is a weakness, since the research protocol for the proposed scoping review follows all the recommendations of the Joanna Brings Institute for scoping reviews.

The methodology was rewritten, indicating the steps proposed by the JBI, the study population, the databases that will be consulted, the mnemonic used, inclusion and exclusion criteria, selection of publications, data extraction and the expected results, in order to provide greater clarity. In addition, the manuscript was reviewed by an English language specialist.

During the review, it was possible to identify the need for restructuring to adopt a global approach. Thus, in order to meet the objective of the review, the paragraph dealing with security policy in Brazil was removed and the text was rewritten.

Considering that the objective of the study is complex, it is understood that the approach proposed to answer the question of this scoping review is adequate, allowing the mapping of the repercussions arising from the phenomenon of the second victim in a broad, exploratory and descriptive way.

Regulated and unregulated terms aligned with professionals with higher and secondary education were included in the text.

In the health sciences descriptor (DeCS), “health professionals” are described as people who work in one of the professions related to health sciences, while “health workers” include, in addition to health professionals, workers who work in health service buildings, workers who work in the administrative area, pantry services, security, reception, cleaning, maintenance and who are not the focus of this study.

The sources are the selected studies/publications. For greater clarity, the term has been changed and restructured within the eligibility criteria. Thus, primary and secondary studies will be included.

A new search was conducted on January 10, 2025

The study will use primary and secondary studies with qualitative and quantitative methodological designs, as well as mixed methods studies. Dissertations, theses and publications from societies, foundations and networks that address the issue of the second victim phenomenon will also be used as gray literature. The use of gray literature aims to identify works published by organizations focused on the study, updating, debate and dissemination in the area of patient safety. This section has been rewritten.

The selection will be led by the corresponding author, this information has been included in the text.

This information that was disconnected from the material has been removed.

The second victim phenomenon causes psychological suffering, and it is important to characterize how this suffering presents itself and what the repercussions of this moment of professional vulnerability are. In this way, this research can contribute to mapping evidence on the phenomenon.

The CAPES system had no role in the study design, data collection and analysis, decision to publish, or preparation of the manuscript. This funding agency represents a government agency that funded the corresponding author's doctoral scholarship with funding code 001 (social demand).

---

## [Decision Letter · Decision Letter 1]

Dear Dr. Vianna Gonçalves da Costa,

We look forward to receiving your revised manuscript.

Kind regards,

Vincenzo De Luca

Academic Editor

PLOS ONE

Reviewers' comments:

Reviewer's Responses to Questions

**Comments to the Author**

1. Does the manuscript provide a valid rationale for the proposed study, with clearly identified and justified research questions?

Reviewer #1: Yes

2. Is the protocol technically sound and planned in a manner that will lead to a meaningful outcome and allow testing the stated hypotheses?

Reviewer #1: Partly

3. Is the methodology feasible and described in sufficient detail to allow the work to be replicable?

Reviewer #1: No

4. Have the authors described where all data underlying the findings will be made available when the study is complete?

Reviewer #1: Yes

5. Is the manuscript presented in an intelligible fashion and written in standard English?

Reviewer #1: Yes

You may also provide optional suggestions and comments to authors that they might find helpful in planning their study.

Reviewer #1: Thank you for revising your manuscript.

Title:

• Consider revising from ‘who went through the second victim phenomenon’ to ‘who have experienced the second victim phenomenon’

Abstract

• You introduce the term second victim phenomenon but don’t define what it is. Defining it can enhance clarity in the abstract.

Introduction

• Specify what patient safety ‘framework’ entails. Are you referring to the second term victim being introduced in the ‘field’ of patient safety? Or a specific framework?

• You cite existing systematic and scoping reviews that relate to the topic. Specifically, you reference one scoping review protocol that aligns with your proposed scoping review but note that you are unable to verify the objectives as the manuscript file is not available. Upon reviewing this registration, the objective is identified under the description section (https://doi.org/10.17605/OSF.IO/5CDMU). As it reads, the objective of this scoping review (“to describe the types of psychological and physical symptoms experienced by healthcare professionals who become second victims after a patient safety incident and its impact on their social and professional lives”) is the same objective you propose in your protocol (“to map and characterize the repercussions experienced by healthcare professionals who have gone through the second victim phenomenon due to patient safety-related incidents in healthcare services”). As a result, your proposed reviews are very similar, except that you are not limiting your search to English-language sources. Consider the value of your review as it may be duplicative.

Eligibility Criteria

• If the search is global, should the search for theses/dissertations be broader and not limited to the Brazilian Digital Library of Theses and Dissertations? Consider using a database such as Proquest Dissertations and Theses Global.

• Exclusion – you indicated “We will also exclude studies incorporated into an evidence synthesis, except in cases where the data presented are not otherwise covered in the synthesis.” This requires further elaboration. What specific process will you implement to operationalize this given that based on your review of existing literature, there is a wide range of evidence syntheses on this topic? How will you manage duplication across evidence syntheses? How will your data extraction and analysis differ based on your inclusion of primary and secondary analysis?

• If theses/dissertations are being included, why are undergraduate theses being excluded?

• Specify what societies, foundations or networks will you review for gray literature. Being explicit is important for transparency.

• Population – “the first group includes nurses, physicians, physical therapists, speech therapists, and others.” Suggest state that these are examples rather than presenting it as a specific list.

• There is no mention of date limiters. Will you be reviewing the literature published within a specific time frame?

Screening and selection

• How many articles or what percent of articles will you use to pilot?

Data analysis

• It is unclear which variables require a mean and median measure.

Expected outcomes

• This section is quite vague and can benefit from a more detailed discussion about potential implications/significance by linking it back to the issue

The Preferred reporting items for systematic review and meta-analysis protocols checklist is missing.

There are outstanding issues that need to be addressed. More importantly, an existing scoping review (based on your review of existing knowledge syntheses and protocols) on the same topic and objective already exists. Unless there is a strong justification for how this review is different than the existing registered scoping protocol in OSF as referenced in the manuscript, this may be duplicative.

**Do you want your identity to be public for this peer review?** For information about this choice, including consent withdrawal, please see our Privacy Policy

Reviewer #1: No

---

## [Author Response · Author response to Decision Letter 2]

23 Mar 2025

We appreciate the contributions to the presentations and understand that they are relevant.

After reading the reviewers' responses, the manuscript was revised to meet the reviewers' recommendations. In addition, it underwent major spelling and language revision.

Therefore, we hope that the revised manuscript meets the guidelines for publication.

Yours sincerely

---

## [Decision Letter · Decision Letter 2]

Dear Dr. Vianna Gonçalves da Costa,

Thank you for submitting your manuscript to PLOS ONE. After careful consideration, we feel that it has merit but does not fully meet PLOS ONE’s publication criteria as it currently stands. Therefore, we invite you to submit a revised version of the manuscript that addresses the points raised during the review process.

We look forward to receiving your revised manuscript.

Kind regards,

Vincenzo De Luca

Academic Editor

PLOS ONE

Journal Requirements:

Reviewers' comments:

Reviewer's Responses to Questions

**Comments to the Author**

1. Does the manuscript provide a valid rationale for the proposed study, with clearly identified and justified research questions?

Reviewer #1: Yes

2. Is the protocol technically sound and planned in a manner that will lead to a meaningful outcome and allow testing the stated hypotheses?

Reviewer #1: No

3. Is the methodology feasible and described in sufficient detail to allow the work to be replicable?

Reviewer #1: No

4. Have the authors described where all data underlying the findings will be made available when the study is complete?

Reviewer #1: Yes

5. Is the manuscript presented in an intelligible fashion and written in standard English?

Reviewer #1: No

You may also provide optional suggestions and comments to authors that they might find helpful in planning their study.

Reviewer #1: Thank you for the opportunity to review the manuscript again. There are still several sections within the manuscript that are unclear which I have identified below. I note that there was no detailed response provided by the authors in response to the previous comments I provided. The response back to the reviewers was generic stating: “We appreciate the contributions to the presentations and understand that they are relevant. After reading the reviewers' responses, the manuscript was revised to meet the reviewers' recommendations. In addition, it underwent major spelling and language revision. Therefore, we hope that the revised manuscript meets the guidelines for publication.”

1. The abstract needs to align with changes in text – based on the revisions, gray literature will also be retrieved from Proquest Dissertations and Theses Global

2. Introduction – “The term second victim is introduced within the patient safety framework and refers to healthcare professionals who experience psychological and professional distress following an adverse patient event” – this was previously raised and it is still not clear what ‘the patient safety framework’ entails. Is this a specific framework or are you speaking in general terms that this has been introduced in the field of patient safety?

3. Eligibility criteria - “Additionally, within gray literature, publications cited in the reference list of the evidence synthesis will be considered if they were published by societies, foundations, or networks addressing the second victim phenomenon” – What does evidence synthesis refer to?

4. Exclusion criteria – “We will exclude abstracts from scientific events, opinion articles, and publications in which the studied population comprises students from either regulated or unregulated professions.” It is not clear why abstracts from scientific events, opinion articles are separated out from publications.

5. Exclusion criteria - Further, in your eligibility criteria, you state: “Given that a scoping review can include all types of literature 26-28, this study will consider primary and secondary studies identified in the databases used in the preliminary search.” Since you state ‘all types of literature’ it is unclear if you are excluding or including other forms of literature beyond empirical studies (e.g. peer-reviewed opinion articles, discussion papers, etc) and gray literature.

6. Exclusion criteria - “Regarding primary sources, we will exclude those incorporated into an evidence synthesis. These publications will be identified in the third stage of the search strategy—while reviewing the reference lists of the studies included in the evidence synthesis.” This is still unclear to me. How will you know which publications are incorporated into evidence synthesis during your initial screening?

7. Search strategy: “To ensure the inclusion of up-to-date evidence, we will perform a final search before presenting the results in the review article. This search will cover the same period and align with the last search described in the protocol. In the second step, an experienced librarian specializing in scoping review search strategies will conduct a comprehensive literature search.” It is unclear why you are conducting a final search and then a comprehensive search. Does the first ‘final search’ pertain to MEDLINE? If MEDLINE is a database that is included in your in search, it should be noted in your abstract as well.

8. Search strategy- In the third step, we will examine the reference lists of the full-text sources selected for inclusion in the review, and publications that meet the inclusion criteria may be included”. Previously you mentioned that you would include a review of publications identified in evidence synthesis but this is not mentioned in your search strategy.

**Do you want your identity to be public for this peer review?** For information about this choice, including consent withdrawal, please see our Privacy Policy

Reviewer #1: No

---

## [Author Response · Author response to Decision Letter 3]

27 May 2025

3. Is the methodology feasible and described in sufficient detail to allow the work to be replicable?

Reviewer #1: No

We appreciate the observation. The text has been improved to meet the observation regarding the reproduction and robustness of the method. We note that the method and materials have been described in detail. That there is no sample calculation because it is a scoping review. The population, the concept and the context were described. The three steps were mentioned in the search strategy. The first stage of the search strategy was presented in the protocol, the second and third stages were described, however the result will be presented in the final article.

5. Is the manuscript presented in an intelligible fashion and written in standard English?

Reviewer #1: No

We justify that after the last revision, points were included to meet the reviewers' needs and, unfortunately, there was no adaptation to the English language. We affirm our commitment to the written form of English and inform that the revised version has undergone corrections and adjustments so that the manuscript is clear in the proposed language.

1. The abstract needs to align with changes in text – based on the revisions, gray literature will also be retrieved from Proquest Dissertations and Theses Global

Thank you for pointing out the inadequacy of the abstract. The abstract was revised and changes in the body of the manuscript were also made to the abstract.

2. Introduction – “The term second victim is introduced within the patient safety framework and refers to healthcare professionals who experience psychological and professional distress following an adverse patient event” – this was previously raised and it is still not clear what ‘the patient safety framework’ entails. Is this a specific framework or are you speaking in general terms that this has been introduced in the field of patient safety?

The term "structure" is being used in general, it refers to the concepts, objectives and aspects related to the theme of patient safety. To improve the text, grammatical adjustments were made in this excerpt

3. Eligibility criteria - “Additionally, within gray literature, publications cited in the reference list of the evidence synthesis will be considered if they were published by societies, foundations, or networks addressing the second victim phenomenon” – What does evidence synthesis refer to?

The synthesis of evidence according to the JBI guideline, are the registries, publications selected in the last stage of screening and eligible for analysis. In these, the reference lists will be analyzed in order to include publications that have not been captured in the search strategy and that

4. Exclusion criteria – “We will exclude abstracts from scientific events, opinion articles, and publications in which the studied population comprises students from either regulated or unregulated professions.” It is not clear why abstracts from scientific events, opinion articles are separated out from publications.

In order to clarify, we reformulated the paragraph. Primary studies, opinion articles and any other publication that has students as its population will be excluded, since the objective of this research is professionals trained and inserted in the context of work.

5. Exclusion criteria - Further, in your eligibility criteria, you state: “Given that a scoping review can include all types of literature 26-28, this study will consider primary and secondary studies identified in the databases used in the preliminary search.” Since you state ‘all types of literature’ it is unclear if you are excluding or including other forms of literature beyond empirical studies (e.g. peer-reviewed opinion articles, discussion papers, etc) and gray literature.

We appreciate the observation, when reviewing the text we found that it was not clear due to the writing. We revised the text to make it explicit that "primary research and reviews will be eligible, provided that they have repercussions resulting from the phenomenon of second victim in health professionals" and "abstracts of scientific events, opinion articles, and publications that did not meet the PCC and the objective of this review will be excluded. Studies on the adaptation and validation of instruments, as well as studies on the prevalence of symptoms, studies on forms of coping and forms of support that do not show the repercussions. In addition to these, publications that are not available in full will also be excluded, as well as those that exclusively addressed students from regulated or non-regulated health professions".

6. Exclusion criteria - “Regarding primary sources, we will exclude those incorporated into an evidence synthesis. These publications will be identified in the third stage of the search strategy—while reviewing the reference lists of the studies included in the evidence synthesis.” This is still unclear to me. How will you know which publications are incorporated into evidence synthesis during your initial screening?

This question made us reflect and return to the manual to discuss the screening and selection process. Thus, this passage was changed based on the recommendations of the method. This stage is the third stage of the search strategy and at this point the reference lists of articles / publications included for review will be analyzed, so that any publication that has not been captured by the search strategy and that is of interest to the research is identified. Revised text: "In order to meet the third stage of the search strategy, the list of references of the selected publications will be examined after the screening carried out in the second stage and following the selection and screening path recommended for the scope reviews, the titles that meet the eligibility criteria will be manually selected. Thus, the abstracts and the full text will be read.

7. Search strategy: “To ensure the inclusion of up-to-date evidence, we will perform a final search before presenting the results in the review article. This search will cover the same period and align with the last search described in the protocol. In the second step, an experienced librarian specializing in scoping review search strategies will conduct a comprehensive literature search.” It is unclear why you are conducting a final search and then a comprehensive search. Does the first ‘final search’ pertain to MEDLINE? If MEDLINE is a database that is included in your in search, it should be noted in your abstract as well.

To clarify this context, we clarify that the text has been revised. Thus, we record that a preliminary search was carried out The Virtual Health Library (VHL), Medical Literature Analysis and Retrieval System (MEDLINE), Cumulative Index to Nursing and Allied Health Literature (CINAHL) and Open Science Framework (OSF), PROSPERO and Joanna Briggs Institute (JBI) databases were found, and no manuscript or protocol record with a similar objective was found. According to the stages of the search strategy, the first stage was carried out on 01/08/2025, with a search limited to a MEDLINE database (PUBMED), and the feasibility of the proposal was verified. For the second stage, which is the survey of the databases, a librarian will be consulted to establish the search strategy. The result of the second stage will be indicated in the presentation of the research article.

8. Search strategy- In the third step, we will examine the reference lists of the full-text sources selected for inclusion in the review, and publications that meet the inclusion criteria may be included”. Previously you mentioned that you would include a review of publications identified in evidence synthesis but this is not mentioned in your search strategy.

The review was carried out as mentioned in the previous items.

---

## [Decision Letter · Decision Letter 3]

Repercussions suffered by health professionals who have experienced the phenomenon of the second victim due to incidents related to patient safety: a scoping review protocol

PONE-D-24-36642R3

Dear Dr. Vianna Gonçalves da Costa,

We’re pleased to inform you that your manuscript has been judged scientifically suitable for publication and will be formally accepted for publication once it meets all outstanding technical requirements.

Kind regards,

Vincenzo De Luca

Academic Editor

PLOS ONE

Additional Editor Comments (optional):

Reviewers' comments:

Reviewer's Responses to Questions

**Comments to the Author**

1. Does the manuscript provide a valid rationale for the proposed study, with clearly identified and justified research questions?

Reviewer #1: Yes

2. Is the protocol technically sound and planned in a manner that will lead to a meaningful outcome and allow testing the stated hypotheses?

Reviewer #1: Yes

3. Is the methodology feasible and described in sufficient detail to allow the work to be replicable?

Reviewer #1: Yes

4. Have the authors described where all data underlying the findings will be made available when the study is complete?

Reviewer #1: Yes

5. Is the manuscript presented in an intelligible fashion and written in standard English?

Reviewer #1: Yes

You may also provide optional suggestions and comments to authors that they might find helpful in planning their study.

Reviewer #1: Thank you for revising your manuscript. You've enhanced clarity and I have some outstanding minor suggestions:

Introduction – “In the JBI database, we identified two registries: the first being a protocol and

the second an article with the aim of synthesizing the best available evidence to

explore the meaning of nurses' experiences as second victims of adverse nursing errors

19-20.” I believe you mean you found two papers, not registries.

Eligibility criteria – “Examples of higher education or graduate professionals include nurses, doctors, physical therapists, speech therapists, and nutritionists, among others. Mid-level or non-licensed professionals include nursing technicians, laboratory technicians, and radiology technicians, among others.” Since the review is global, suggest making a minor change as in some countries, laboratory technicians, radiology technicians etc. are licensed professionals.

Exclusion criteria: This section is written in the past tense – suggest to ensure consistency throughout your manuscript

Expected results – suggest clarifying if you mean ‘causes psychological suffering’ versus ‘causes psychic suffering’

I suggest some additional support to address grammatical issues throughout the manuscript for readability and accuracy.

Good luck with your review. Thanks again for the opportunity to provide feedback.

**Do you want your identity to be public for this peer review?** For information about this choice, including consent withdrawal, please see our Privacy Policy

Reviewer #1: No

---

## [Editor Report · Acceptance letter]

PONE-D-24-36642R3

PLOS ONE

Dear Dr. Vianna Gonçalves da Costa,

I'm pleased to inform you that your manuscript has been deemed suitable for publication in PLOS ONE. Congratulations! Your manuscript is now being handed over to our production team.

Kind regards,

on behalf of

Dr. Vincenzo De Luca

Academic Editor

PLOS ONE